# Trends and Motivations in Dietary Supplement Use Among People with Diabetes: A Population-Based Analysis Using National Health and Nutrition Examination Survey Data from the 2009–2020 Period

**DOI:** 10.3390/nu16234021

**Published:** 2024-11-24

**Authors:** Yan Jiang, Xuhui Chen, Zhen Cai, Ying Yao, Shuaiwen Huang

**Affiliations:** 1Nursing Department, Tongji Hospital, Tongji Medical College, Huazhong University of Science and Technology, 1095 Jiefang Avenue, Wuhan 430030, China; 2Department of Ophthalmology, Tongji Hospital, Tongji Medical College, Huazhong University of Science and Technology, 1095 Jiefang Avenue, Wuhan 430030, China; 3Department of Nutrition, Tongji Hospital, Tongji Medical College, Huazhong University of Science and Technology, 1095 Jiefang Avenue, Wuhan 430030, China; 4Division of Nephrology, Tongji Hospital, Tongji Medical College, Huazhong University of Science and Technology, 1095 Jiefang Avenue, Wuhan 430030, China; 5Department of General Practice, Tongji Hospital, Tongji Medical College, Huazhong University of Science and Technology, 1095 Jiefang Avenue, Wuhan 430030, China

**Keywords:** dietary supplement use, diabetes, trends, motivations, NHANES

## Abstract

Objectives: Dietary supplements have gained attention among people with diabetes as an alternative and complementary treatment, yet there is a limited understanding of supplement use and the motivations behind it. Methods: Data from the National Health and Nutrition Examination Survey (NHANES) from the 2009–2020 period were analyzed, including data on 5784 people with diabetes aged 20 years or older. Dietary supplement use was self-reported. Trends in supplement use were examined across three periods: 2009–2012, 2013–2016, and 2017–2020. Statistical analyses were conducted while considering NHANES’s complex survey design to provide nationally representative estimates for the general noninstitutionalized population of the United States. Results: A total of 61.72% of individuals reported using dietary supplements with a notable increase over time. Supplement users were generally older, more likely to be female, better educated, and had superior blood glucose control with lower smoking rates compared to non-users. Common supplements included multivitamins, multimineral supplements, vitamin D, calcium, zinc, vitamin C, and fish oil. Only 44.58% of individuals used dietary supplements based on medical advice, with the rest opting for self-directed usage. The primary specific health reasons for supplement use were to improve bone health and heart health and enhance the immune system. Conclusions: Dietary supplement use is prevalent among people with diabetes, and most diabetic supplement use is self-directed, which reflects a growing trend toward complementary therapies. Healthcare providers are encouraged to inquire about patients’ use of supplements and offer appropriate guidance as an integral component of comprehensive diabetes management.

## 1. Introduction

Diabetes is a global public health issue with a rising incidence and prevalence. According to the International Diabetes Federation, over 400 million people worldwide have diabetes, and this number is expected to reach 700 million by 2045 [1,2]. Diabetes significantly impacts patients’ health and imposes a substantial burden on society and the economy. Despite advancements in diabetes management, there remains a substantial number of patients whose needs are not fully met through conventional medical approaches alone.

Against this backdrop, dietary supplements have gained attention among people with diabetes as an alternative and complementary treatment [3]. These supplements, including vitamins, minerals, herbal extracts, and other functional food ingredients, are believed to offer benefits such as blood glucose regulation, antioxidant properties, and anti-inflammatory effects [4,5]. However, scientific evidence of their effectiveness and safety remains inconclusive, and the usage patterns, motivations, and influencing factors among patients are not well understood.

To address this gap, we used the latest data from the National Health and Nutrition Examination Survey (NHANES) to (1) describe the overall use of dietary supplements among people with diabetes, (2) compare the characteristics of supplement users and non-users, (3) identify the most commonly used supplements by those managing diabetes, (4) evaluate the efficacy of these supplements by comparing their effects on fasting glucose and HbA1c levels, (5) determine the differences in demographic and health-related characteristics between doctor-advised and self-directed supplement use, and (6) assess the motivations and reasons for supplement use.

This study aims to investigate the current status of dietary supplement use among people with diabetes. By understanding usage patterns, reasons for use, efficacy evaluations, and information sources, we hope to provide insights that can help healthcare professionals better guide patients in the rational use of dietary supplements, thereby improving overall diabetes management outcomes.

## 2. Materials and Methods

### 2.1. Study Population

The data for this study come from the NHANES, a nationwide, cross-sectional survey conducted over multiple years in the United States. The NHANES uses a multistage, stratified probability sampling design to collect demographic, dietary, examination, laboratory, and questionnaire data, providing nationally representative estimates for the adult population in the U.S. [6].

NHANES participants provided informed consent to participate in this study, and the survey design was approved by the National Center for Health Statistics Ethics Review Board. This study utilized publicly available anonymized data from the NHANES; therefore, no further institutional review board approval was required.

In this analysis, we included people with diabetes from the continuous NHANES in the 2009–2020 period. Only those who were aged 20 years or older at the time of the baseline survey were included. Subjects lacking supplement use data were excluded. As a result, 5784 people were left in our cohort for analysis (Figure 1).

### 2.2. Definitions of Diabetes and Its Complications

Given the high number of missing values in the laboratory tests, we used self-reported data to confirm diabetes. The subjects were asked whether they had been diagnosed with diabetes by a doctor in the past, and if the answer was yes, they were considered people with diabetes. In addition, based on the ADA’s classification and diagnosis of diabetes, the following factors were used to confirm diabetes: meeting (i) a fasting blood glucose (FPG) ≥ 126 mg/dL (7.0 mmol/L), (ii) a 2 h blood glucose (2 h PG) ≥ 200 mg/dL (11.1 mmol/L) during the oral glucose tolerance test (OGTT), and (iii) HbA1c ≥ 6.5% (48 mmol/mol) [7].

Participants who self-reported diabetes were asked to disclose the age at which they were first diagnosed with diabetes mellitus (DM), as well as their current use of medications to manage their blood sugar levels, including diabetic pills and insulin. Diabetic retinopathy (DR) is classified as someone who answered “yes” to the statement “Diabetes affects the eyes/has retinopathy”. Diabetic nephropathy (DN) is defined as a condition in individuals with diabetes characterized by a urinary albumin-to-creatinine ratio exceeding 30 mg/g or an estimated glomerular filtration rate (eGFR) below 60 mL/min/1.73 m^2^. The CKD-EPI (2021) equation was used to estimate the GFR using the following formula [8]: eGFR = 142 × [min(serum creatinine in mg/dL)/κ, 1)]**^α^ × [max(serum creatinine/κ, 1)]^** − 1.20^ × 0.9938^**age^ × (1.012 if female).

### 2.3. Definition of Covariates

The baseline survey questionnaire was used to collect covariate information, including age, sex, race/ethnicity, educational level, smoking status, and drinking habits. Additionally, the baseline questionnaire recorded a history of diabetes or heart disease in the family, self-reported health status, and baseline self-reported medical history of hypertension, cardiovascular diseases, chronic obstructive pulmonary disease, and cancer. Height, weight, and blood pressure were measured at the mobile examination center, BMI was calculated using weight (kg)/height (m)^2^, while blood pressure was obtained by averaging multiple measurements. Data on blood lipids, blood sugar, blood creatinine, and urine albumin-to-creatinine ratio were derived from laboratory tests. Detailed testing methods and specifics can be found on the NHANES’s official website.

### 2.4. Dietary Supplement Data

The NHANES collected self-reported information on dietary supplement use during in-home interviews conducted by trained interviewers using a computer-assisted interview system. Individuals were asked if they indicated they had taken any dietary supplements in the past 30 days and were asked by the interviewer to view the supplements, if any. Supplement users were asked whether they took the product on their own or under the advice of a doctor. The dietary supplement interview also included questions about the reasons for taking supplements. More specifically, participants were shown a hand card with a list of potential reasons and were asked to select one or more reasons for each supplement. Referring to the previous literature [9], we manually reviewed all dietary supplements in the NHANES dataset and categorized them by name.

### 2.5. Statistical Analysis

We accounted for the complex survey design factors of the NHANES, including sample weights, clustering, and stratification, as specified in the instructions for utilizing NHANES data. To compare baseline characteristics, we employed the Rao-Scott χ^2^ test for categorical variables and an analysis of variance adjusted for sampling weights for continuous variables. Adjusted utilization estimates for dietary supplements were calculated based on the NHANES in the 2009–2020 period. A trend analysis was performed by combining data from three 4-year periods: 2009–2012, 2013–2016, and 2017–2020. Logistic regression was employed to assess trends in dietary supplement usage over time. A two-tailed *p* value of less than 0.05 was considered to be statistically significant; all statistical analyses were performed using SAS 9.4, which allows for the appropriate use of the NHANES weights to project the results of the analysis to the general noninstitutionalized population in the U.S.

## 3. Results

### 3.1. Utilization of Dietary Supplements

Our study encompassed 5784 people with diabetes, with 3294 reporting the use of dietary supplements, representing 56.95% of the cohort, and an adjusted rate of utilization of 61.72%. A data analysis spanning from 2009 to 2020 revealed a consistent upward trend in dietary supplement usage among individuals with diabetes, with a significant time trend (*p* < 0.0001). Usage rates increased from 54.53% in the 2009–2012 period to 67.94% in the 2017–2020 period, as detailed in Table 1 and illustrated in Figure 2. This trend was consistently observed across all analyzed demographic and health-related characteristics, highlighting that the widespread integration of supplements into the health management routines of individuals with diabetes is now a reality.

### 3.2. Characteristics of Supplement Users vs. Non-Users

A comparative analysis of the baseline characteristics, as detailed in Table 2, indicates that individuals with diabetes who use dietary supplements are typically older, predominantly female, more educated, and exhibit better glycemic control, evidenced by lower levels of glycated hemoglobin and fasting blood glucose. These individuals also have lower levels of low-density lipoprotein (LDL) cholesterol and total cholesterol, along with reduced smoking rates. Predominantly, these patients are non-Hispanic white and report a better self-assessed health status. Although the prevalence of diabetic retinopathy remains similar between supplement users and non-users, the incidence of diabetic nephropathy is notably higher among supplement users. Furthermore, a higher proportion of individuals in the supplement group report comorbid conditions such as hypertension, cardiovascular diseases, and cancer. While the use of insulin and hypoglycemic drugs shows no significant difference between the two groups, the use of lipid-lowering and antihypertensive medications is more prevalent among those who take dietary supplements.

Figure 3 provides a comparison of fasting blood glucose and glycated hemoglobin, categorized according to specific types of dietary supplements (top 20 commonly used supplements). When we grouped patients by specific types of supplements, there were minimal differences in the fasting blood glucose levels between most groups. However, the HbA1c levels were lower in the subjects who took folic acid, lutein, and magnesium.

### 3.3. Types and Reasons for Dietary Supplement Use

As shown in Figure 4A, the most commonly used supplements included multivitamins, multimineral supplements, vitamin D, calcium, zinc, vitamin C, and fish oil. The primary specific health reasons for using these supplements, as elucidated in Figure 4B, were to improve overall health, maintain wellness, support bone health, augment their diet, boost energy levels, enhance heart health, and strengthen immune function. Notably, controlling blood sugar was not a major motive for supplement use among these patients. Appendix A presents a ranking of the reasons for using dietary supplements (top 10 commonly used supplements).

### 3.4. Doctor-Advised vs. Self-Directed Supplement Use

Only 44.58% of people with diabetes used dietary supplements based on medical advice, with the rest opting for self-directed usage. Those following a doctor’s recommendations were generally older, in poorer health, and more likely to suffer from comorbid conditions such as cancer, hypertension, and heart disease, as outlined in Table 3. Individuals who were taking dietary supplements based on a physician’s advice exhibited a higher incidence of diabetic nephropathy, lower levels of glycated hemoglobin, more frequent use of lipid-lowering medications, and lower total cholesterol levels. Appendix A shows the most commonly used types of supplements and reasons for dietary supplement use under doctor-advised and self-directed conditions.

## 4. Discussion

The findings of this study underscore the growing reliance on dietary supplements among patients with diabetes, particularly as a complementary strategy to conventional diabetes management. Our analysis, based on data from the NHANES in the 2009–2020 period, reveals that a significant proportion of people with diabetes, specifically 61.72%, use dietary supplements, and this trend has been increasing over the years. This rise in dietary supplement use aligns with the global trend of patients seeking alternative therapies to manage chronic conditions like diabetes [10,11,12]. Moreover, the similarity in trends across different baseline characteristics such as age, gender, and education level suggests that the increase in dietary supplement use is a widespread phenomenon among people with diabetes. This trend has also been observed in other studies [13,14], which have suggested that the growing popularity of dietary supplements may be driven by increased awareness of the potential health benefits of these products and the rise of integrative medicine. This widespread adoption could also be attributed to the perceived benefits of dietary supplements, including improved antioxidant properties and anti-inflammatory effects [15,16]. However, it is important to note that while the popularity of these supplements is growing, the scientific evidence supporting their efficacy and safety remains mixed [17,18,19,20]. Healthcare professionals should be aware of this trend and guide their patients accordingly, ensuring that any supplement use complements rather than complicates their diabetes management plan.

Our study also highlights important demographic distinctions between supplement users and non-users. People with diabetes who use dietary supplements tend to be older and more educated and are more likely to be female and white. According to the previous literature, it is estimated that 57.6% of Americans consume dietary supplements, with higher proportions being observed among older adults and women [12]. These characteristics are similar to those of our study’s population. People with diabetes who use dietary supplements also exhibit better blood glucose control and report better overall health, with a lower prevalence of smoking. This demographic pattern suggests that certain groups may be more inclined to adopt supplementary treatments, possibly due to greater health awareness or access to health information [21,22].

We found that people with diabetes who used dietary supplements had better glycemic control, reflected in lower fasting blood glucose and HbA1c levels, even though there were no significant differences in insulin and oral hypoglycemic agent use between those who used supplements and those who did not. When we grouped subjects by specific types of supplements, there were minimal differences in fasting blood glucose levels between most groups. However, the HbA1c levels were lower in subjects who took folic acid, lutein, and magnesium. Consistent with previous studies [23,24,25], supplementation with lutein, folic acid, and magnesium is thought to have potential benefits for insulin resistance and glycemic control. Studies indicate that folic acid can enhance endothelial function and reduce inflammation levels, both of which are crucial for improving insulin signaling [26,27]. Folic acid indirectly increases insulin sensitivity by lowering levels of homocysteine, an amino acid associated with insulin resistance [28]. Lutein, an antioxidant, has been shown to reduce oxidative stress and inflammation, key drivers of insulin resistance [29]. Additionally, lutein can improve the function of adipocytes, thereby aiding in the enhancement of insulin sensitivity [30]. Magnesium supplementation directly boosts the activity of insulin receptors, thus improving insulin sensitivity and glycemic control [31]. However, due to the cross-sectional design of our study, it is difficult to establish a causal relationship. Future prospective studies are needed to further identify and confirm supplements that may have hypoglycemic effects.

The 20 most commonly used supplements among people with diabetes, as identified in our study, include multivitamins, multimineral supplements, vitamin D, calcium, zinc, vitamin C, fish oil, and others. These supplements are popular not only for their potential to support overall health but also for their specific benefits related to diabetes management, such as improved bone health and heart health [32,33,34,35,36]. The use of omega fatty acids, iron, magnesium, and glucosamine further indicates a focus on managing the broader health challenges associated with diabetes.

Interestingly, only 44.58% of people with diabetes actually use dietary supplements under the advice of doctors, and most individuals use dietary supplements by themselves. The characteristics of those who take dietary supplements based on a doctor’s recommendation differ from those who take them independently. The former group tends to be older, is more likely to be female, and reports poorer health with a higher prevalence of serious conditions such as cancer, hypertension, and heart disease. This suggests that doctors may be recommending supplements to patients with more complex health needs, possibly as a way to mitigate the effects of these comorbidities or to enhance the overall treatment plan. Finally, our study identifies the primary reasons for dietary supplement use among patients with diabetes as improving overall health, maintaining wellness, supporting bone health, augmenting their diet, boosting energy levels, enhancing heart health, and strengthening immune function. These motivations appear closely linked to the common complications of diabetes, such as cardiovascular disease, bone demineralization, and weakened immunity. For example, cardiovascular issues, a frequent complication of diabetes, may explain why many patients prioritize heart health when selecting supplements [37]. Similarly, the focus on immune function could be attributed to the increased susceptibility to infections observed in diabetic populations [38]. Moreover, the diverse range of reasons for supplementation may also reflect varying patient perceptions about disease management and health optimization. While some patients may focus on addressing immediate health challenges, such as fatigue or immune support, others may be motivated by preventative strategies, aiming to reduce the long-term impact of diabetes-related complications. These patterns underscore the importance of personalized patient education.

This study offers several insights for clinical practice and public health policy. Firstly, healthcare providers should be aware of the high prevalence of supplement use among people with diabetes and routinely inquire about their use during clinical consultations. This is particularly crucial given the potential interactions between supplements and prescription medications or their impact on disease progression. Clinicians should also provide evidence-based guidance on the use of dietary supplements, helping patients make informed decisions about their health. Secondly, public health initiatives should aim to enhance awareness of the potential risks and benefits of dietary supplements, especially among populations more likely to use these products without medical supervision. Lastly, future research should focus on understanding the motivations behind supplement use among patients with diabetes and explore the long-term health outcomes associated with their use. Large-scale randomized controlled trials are needed to ascertain the efficacy and safety of commonly used supplements in these populations and identify any potential risks associated with their use.

### Strengths and Limitations

One of the major strengths of our study is the use of data from the NHANES, which provides a large, nationally representative sample of the U.S. population. This allows our findings to be generalized to a broader sample of people with diabetes, enhancing the external validity of the results. Furthermore, our study spans a decade (2009–2020), enabling us to capture trends over time and observe changes in dietary supplement use patterns.

However, our study also has limitations that should be acknowledged. First, the cross-sectional design of the NHANES prevents us from drawing causal inferences about the relationship between dietary supplement use and health outcomes. Longitudinal studies would be necessary to determine whether the observed supplement use leads to improved health outcomes over time. Second, our reliance on self-reported data for both diabetes status and dietary supplement use introduces the possibility of recall bias or misreporting. This could lead to inaccuracies in the estimation of supplement use prevalence and the identification of patients with diabetes. Additionally, while we categorized dietary supplements based on available data, we were unable to account for variations in the quality, dosage, and formulation of these supplements, which could influence their effectiveness and safety. Future research should consider these factors, particularly in relation to the potential interactions between dietary supplements and prescribed diabetes medications. Furthermore, this study only included participants diagnosed with diabetes, and thus, it is not appropriate to infer that individuals with diabetes are increasingly relying on supplements compared to those without diabetes.

## 5. Conclusions

Our analysis of NHANES data from 2009 to 2020 indicates an increasing trend in dietary supplement use among individuals with diabetes. This rise reflects a proactive approach by patients to enhance health outcomes and manage their condition more effectively. However, it also prompts important questions about how these supplements are integrated into conventional treatment regimens. Dietary supplement users are generally older, better educated, and exhibit superior glycemic control, indicating targeted use among those with greater health literacy. The most commonly used supplements include multivitamins, multimineral supplements, vitamin D, calcium, zinc, vitamin C, and fish oil, which are chosen for their general and diabetes-specific health benefits. Supplements such as folic acid and magnesium are beneficial for glycemic control, although further research is required to confirm these effects. Significantly, only 44.58% of supplement use is doctor-advised, with the majority being self-directed. This underlines the importance of healthcare providers playing an active role in guiding supplement use to ensure it is safe and effective. Healthcare providers should play a crucial role in advising patients on the appropriate use of dietary supplements, ensuring their use is grounded in sound scientific evidence and tailored to the individual’s overall health status and treatment goals. Further research is necessary to provide clearer guidance on the efficacy and safety of dietary supplements in the context of diabetes management.

## Figures and Tables

**Figure 1 nutrients-16-04021-f001:**
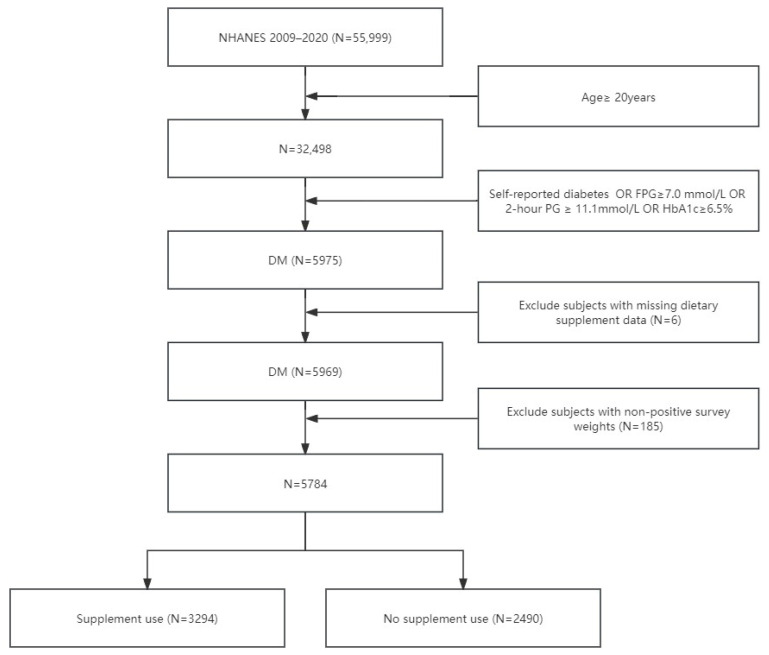
A flow chart illustrating the inclusion and exclusion of study participants. Abbreviations: DM: Diabetes Mellitus; NHANES: National Health and Nutrition Examination Survey; FPG: Fasting Plasma Glucose; HbA1c: Hemoglobin A1c; 2 h PG: 2 h Postprandial Glucose.

**Figure 2 nutrients-16-04021-f002:**
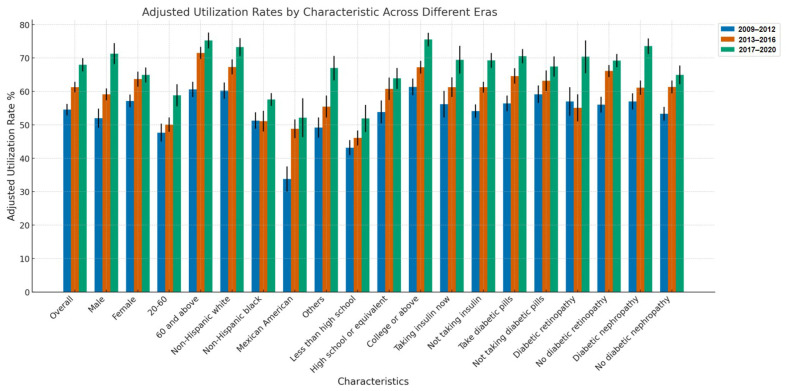
Adjusted utilization rate trends by group and era (2009–2020).

**Figure 3 nutrients-16-04021-f003:**
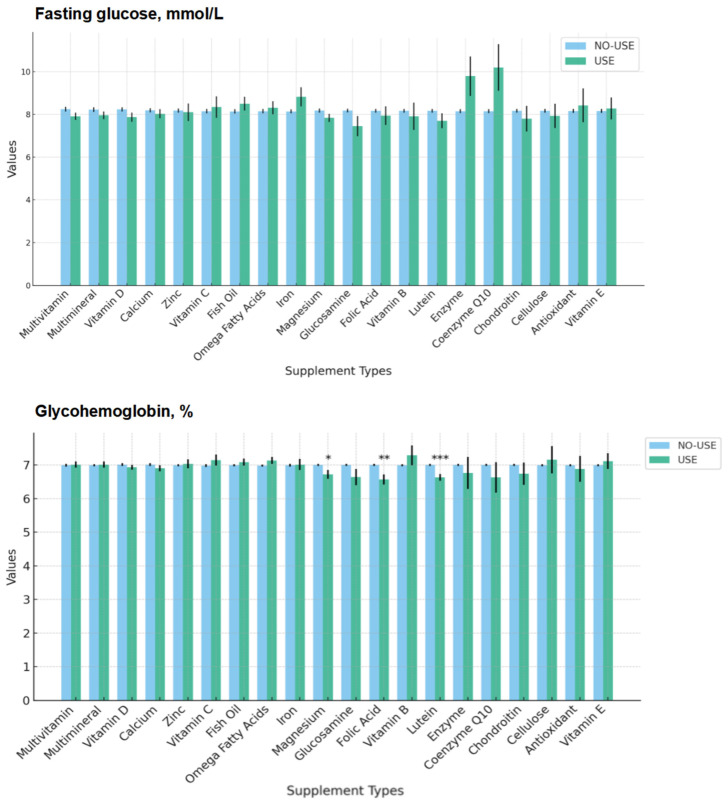
Comparison of fasting blood glucose and glycated hemoglobin, categorized according to specific types of dietary supplements (top 20 commonly used supplements).

**Figure 4 nutrients-16-04021-f004:**
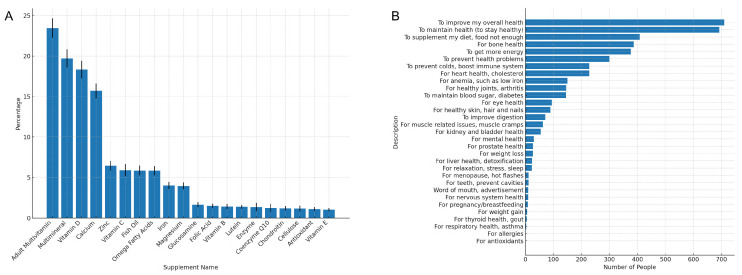
(**A**) Commonly used dietary supplements ordered by frequency of use among people with diabetes. (**B**) Reasons for taking dietary supplements among people with diabetes.

**Table 1 nutrients-16-04021-t001:** Adjusted utilization rate of supplement use based on selected characteristics in U.S. adults with diabetes *.

Characteristics	Adjusted Utilization Rate % (SE)			
	NHANES 2009–2020 (Total)	NHANES 2009–2012 (era1)	NHANES 2013–2016 (era2)	NHANES 2007–2020 (era3)
Overall	61.72 (1.04)	54.53 (1.74)	61.28 (1.58)	67.94 (1.99)
Sex				
Male	64.42 (1.47)	52.00 (2.87)	59.13 (1.79)	71.31 (3.14)
Female	59.24 (1.30)	57.12 (1.91)	63.68 (2.23)	64.93 (2.25)
Age groups (years)				
20–60	52.42 (1.60)	47.67 (2.71)	50.05 (2.13)	58.84 (3.29)
60 and above	69.77 (1.33)	60.62 (2.27)	71.54 (1.79)	75.27 (2.33)
Ethnicity				
Non-Hispanic white	67.36 (1.44)	60.23 (2.44)	67.34 (2.22)	73.28 (2.63)
Non-Hispanic black	53.35 (1.46)	51.26 (2.47)	51.10 (3.07)	57.58 (1.91)
Mexican American	45.66 (2.67)	33.80 (3.75)	48.82 (2.80)	52.13 (5.84)
Others	58.64 (2.09)	49.16 (3.01)	55.45 (3.28)	67.01 (3.65)
Education				
Less than high school	46.61 (1.65)	43.18 (2.24)	46.06 (2.23)	51.94 (4.05)
High school or equivalent	60.31 (1.89)	53.86 (3.43)	60.78 (3.33)	63.89 (3.15)
College or above	68.62 (1.21)	61.37 (2.51)	67.21 (1.92)	75.53 (2.03)
Taking insulin now				
Yes	63.03 (2.14)	56.20 (3.97)	61.27 (2.98)	69.47 (4.17)
No	61.18 (1.14)	54.11 (1.97)	61.26 (1.64)	69.27 (2.25)
Taking diabetic pills to lower blood sugar				
Yes	64.66 (1.32)	56.39 (2.35)	64.61 (2.32)	70.55 (2.16)
No	63.76 (1.73)	59.16 (2.64)	63.22 (3.05)	67.48 (3.03)
Diabetic retinopathy				
Yes	61.75 (2.73)	56.98 (4.29)	55.07 (4.07)	70.37 (4.92)
NO	64.48 (1.18)	56.01 (2.36)	66.10 (1.83)	69.26 (1.99)
Diabetic nephropathy				
Yes	64.65 (1.37)	56.99 (2.46)	61.13 (2.16)	73.54 (2.29)
NO	60.25 (1.35)	53.32 (2.07)	61.35 (1.93)	64.93 (2.79)

* All estimates accounted for complex survey designs and sampling weights of NHANES. Abbreviations: SE: standard error.

**Table 2 nutrients-16-04021-t002:** Characteristics of U.S. adults with diabetes based on NHANES. Overall results and results stratified by supplement use.

	Overall DM Population	Supplement Use	No Supplement Use	*p* Value *
Participants †	5784	3294	2490	
Mean (95% CI) age, years	59.77 (59.16–60.37)	62.16 (61.35–62.98)	55.90 (55.07–56.73)	<0.0001
Women	2783 (46.38)	1682 (50.02)	1101 (44.55)	0.0059
Mean (95% CI) body mass index	32.98 (32.66–33.30)	32.70 (32.29–33.10)	33.44 (32.94–33.94)	0.0233
Ethnicity				<0.0001
Non-Hispanic white	1868 (59.04)	1206 (64.43)	662 (50.34)	
Non-Hispanic black	1525 (14.19)	831 (12.27)	694 (17.30)	
Mexican American	979 (10.12)	454 (7.48)	525 (14.37)	
Others	1412 (16.63)	803 (15.80)	609 (17.97)	
Education				<0.0001
Less than high school	1846 (21.40)	839 (16.16)	1007 (29.84)	
High school or equivalent	1347 (26.42)	753 (25.85)	594 (27.42)	
College or above	2578 (52.14)	1695 (57.97)	883 (42.72)	
Alcohol drinking	3800 (77.13)	2217 (78.68)	1583 (74.55)	0.0069
Smoking status				<0.0001
Never smoked	2995 (50.04)	1750 (50.36)	1254 (49.51)	
Former smoker	1861 (34.30)	1156 (38.26)	705 (27.91)	
Current smoker	923 (15.65)	385 (11.36)	538 (22.57)	
Family history of diabetes or heart attack	3930 (68.74)	2280 (69.05)	1650 (68.25)	0.6900
Self-reported health				<0.0001
Very good to excellent	1005 (21.14)	636 (23.18)	369 (17.84)	
Good	2138 (41.06)	1270 (42.56)	868 (38.64)	
Poor to fair	2636 (37.78)	1386 (34.24)	1250 (43.50)	
Self-reported chronic diseases				
Hypertension	3827 (65.60)	2321 (70.13)	1506 (58.30)	<0.0001
Cardiovascular diseases	1457 (24.89)	896 (26.52)	561 (22.26)	0.0090
Chronic obstructive pulmonary diseases	451 (7.69)	254 (7.76)	197 (7.58)	0.8609
Cancer ‡	730 (14.14)	477 (16.21)	253 (10.79)	0.0003
Diabetic retinopathy	866 (18.61)	508 (17.96)	358 (19.76)	0.3063
Diabetic nephropathy	2156 (33.44)	1255 (35.03)	901 (30.88)	0.0191
Mean (95% CI) fasting glucose, mmol/L	8.45 (8.28–8.61)	8.16 (7.97–8.36)	8.89 (8.55–9.24)	<0.0001
Mean (95% CI) glycohemoglobin, %	7.19 (7.12–7.25)	6.99 (6.92–7.07)	7.50 (7.39–7.61)	0.0008
Mean (95% CI) duration of diabetes, years	6.56 (4.55–8.58)	7.71 (5.53–9.90)	4.52 (0.43–8.62)	0.1781
Taking insulin now	1237 (23.02)	728 (23.55)	509 (22.16)	0.4392
Taking diabetic pills to lower blood sugar	3139 (64.75)	1909 (65.07)	1230 (64.17)	0.6783
Mean (95% CI) systolic pressure, mmHg	128.45 (127.02–129.88)	128.92 (127.09–130.74)	127.45 (125.24–129.65)	0.3367
Mean (95% CI) diastolic pressure, mmHg	74.91 (73.90–75.91)	74.49 (73.28–75.70)	75.81 (74.72–76.90)	0.0732
Taking prescription for HBP	3661 (95.66)	2250 (96.87)	1411 (93.31)	0.0037
Mean (95% CI) direct HDL cholesterol, mmol/L	1.22 (1.21–1.24)	1.24 (1.22–1.26)	1.19 (1.16–1.21)	<0.0001
Mean (95% CI) total cholesterol, mmol/L	4.72 (4.67–4.77)	4.66 (4.59–4.72)	4.82 (4.74–4.90)	0.0006
Mean (95% CI) LDL cholesterol, mmol/L	2.69 (2.63–2.75)	2.62 (2.55–2.69)	2.79 (2.71–2.87)	0.0003
Mean (95% CI) triglyceride, mmol/L	1.77 (1.67–1.87)	1.74 (1.61–1.88)	1.82 (1.71–1.94)	0.3281
Taking prescription for cholesterol	3285 (66.30)	2090 (69.72)	1195 (59.95)	<0.0001
Mean (95% CI) eGFR, mL/min/1.73 m^2^	86.30 (85.46–87.15)	83.90 (82.70–85.10)	90.18 (89.00–91.37)	<0.0001

* The *p* value represents the comparison between supplement users and those not using supplements. † Totals of 13, 138, 463,5, 5, 15, 389, 1025, 972, 1960, 188, 2718, 1561, 342, 342, 311, 311, 2903, 2821, and 1596 participants had missing information for baseline education level, BMI, alcohol drinking, smoking status, self-reported health, taking insulin, taking diabetic pills, taking prescription for cholesterol, taking prescription for HBP, glycohemoglobin, fasting glucose, duration of diabetes, systolic pressure, diastolic pressure, direct hdl cholesterol, total cholesterol, ldl cholesterol, triglyceride, and diabetic retinopathy, respectively. ‡ Skin cancer was not included. Abbreviations: DM: Diabetes Mellitus; CI: Confidence Interval; HBP: High Blood Pressure; LDL: Low-Density Lipoprotein; HDL: High-Density Lipoprotein; eGFR: Estimated Glomerular Filtration Rate.

**Table 3 nutrients-16-04021-t003:** Characteristics of adults with diabetes who take dietary supplements, categorized as self-directed versus doctor-advised use.

	Self-Directed Use	Doctor-Advised Use	*p* Value *
Participants †	1579	1249	
Mean (95% CI) age, years	60.40 (59.33–61.46)	64.60 (63.64–65.57)	<0.0001
Women	743 (45.56)	693 (55.27)	<0.0001
Mean (95% CI) body mass index	32.54 (32.02–33.06)	32.82 (32.29–33.36)	0.4065
Ethnicity			0.2273
Non-Hispanic white	548 (62.43)	468 (65.87)	
Non-Hispanic black	397 (12.46)	318 (11.93)	
Mexican American	215 (7.71)	181 (7.58)	
Others	419 (17.37)	282 (14.60)	
Education			0.0609
Less than high school	365 (14.80)	328 (16.03)	
High school or equivalent	349 (23.97)	295 (28.46)	
College or above	862 (61.22)	624 (55.50)	
Alcohol drinking	1075 (78.25)	844 (79.58)	0.4926
Smoking status			0.4230
Never smoked	852 (52.87)	670 (48.98)	
Former smoker	550 (36.39)	438 (39.23)	
Current smoker	175 (10.72)	141 (11.77)	
Family history of diabetes or heart attack	1071 (68.57)	878 (68.69)	0.9616
Self-reported health			0.0005
Very good to excellent	365 (26.14)	201 (19.69)	
Good	661 (45.96)	459 (41.30)	
Poor to fair	551 (27.88)	589 (38.99)	
Self-reported chronic diseases			
Hypertension	1020 (63.05)	942 (77.55)	<0.0001
Cardiovascular diseases	334 (19.66)	405 (32.64)	<0.0001
Chronic obstructive pulmonary diseases	112 (7.65)	93 (7.50)	0.9261
Cancer ‡	184 (12.95)	229 (20.78)	<0.0001
Diabetic retinopathy	205 (18.02)	225 (17.94)	0.9772
Diabetic nephropathy	522 (31.15)	528 (38.15)	0.0077
Mean (95% CI) duration of diabetes, years	6.81 (3.92–9.70)	9.46 (6.03–12.89)	0.2589
Mean (95% CI) fasting glucose, mmol/L	8.27 (8.01–8.53)	8.01 (7.70–8.31)	0.1698
Mean (95% CI) glycohemoglobin, %	7.08 (6.98–7.18)	6.89 (6.78–7.01)	0.0082
Taking insulin now	297 (23.32)	314 (25.00)	0.5852
Taking diabetic pills to lower blood sugar	869 (62.56)	756 (67.03)	0.1281
Mean (95% CI) systolic pressure, mmHg	128.52 (126.59–130.46)	128.14 (125.32–130.97)	0.8201
Mean (95% CI) diastolic pressure, mmHg	75.02 (73.58–76.46)	72.99 (71.17–74.80)	0.0960
Taking prescription for HBP	988 (96.65)	918 (97.25)	0.6813
Mean (95% CI) direct HDL cholesterol, mmol/L	1.24 (1.21–1.28)	1.25 (1.22–1.28)	0.8616
Mean (95% CI) total cholesterol, mmol/L	4.71 (4.63–4.79)	4.55 (4.44–4.67)	0.0208
Mean (95% CI) LDL cholesterol, mmol/L	2.67 (2.58–2.76)	2.54 (2.43–2.65)	0.0604
Mean (95% CI) triglyceride, mmol/L	1.78 (1.57–1.99)	1.61 (1.51–1.70)	0.1095
Taking prescription for cholesterol	928 (64.61)	867 (75.25)	0.0029
Mean (95% CI) eGFR, mL/min/1.73 m^2^	87.75 (86.17–89.33)	79.39 (77.55–81.22)	<0.0001

* The *p* value represents the comparison between supplement users and those not using supplements. † Totals of 5, 59, 185, 2, 5, 2, 196, 395, 83, 1304, 698, 160, 160, 135, 135, 1380, 1345 and 711 participants had missing information for baseline education level, BMI, alcohol drinking, smoking status, self-reported health, taking insulin, taking diabetic pills, glycohemoglobin, fasting glucose, duration of diabetes, systolic pressure, diastolic pressure, direct hdl cholesterol, total cholesterol, ldl cholesterol, triglyceride, and diabetic retinopathy, respectively. ‡ Skin cancer was not included. Abbreviations: DM: Diabetes Mellitus; CI: Confidence Interval; HBP: High Blood Pressure; LDL: Low-Density Lipoprotein; HDL: High-Density Lipoprotein; eGFR: Estimated Glomerular Filtration Rate.

## Data Availability

The data that support the findings of this study are available from NHANES [https://www.cdc.gov/nchs/nhanes], accessed on 1 June 2024. Furthermore, the cleaned datasets that were analyzed in the current study are also available from the corresponding author upon reasonable request.

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
