# Peer review of "Trends and Motivations in Dietary Supplement Use Among People with Diabetes: A Population-Based Analysis Using National Health and Nutrition Examination Survey Data from the 2009–2020 Period"

_nutrients, 2024, doi:10.3390/nu16234021_

Round 1
Reviewer 1 Report
Comments and Suggestions for Authors
Overall, a pleasure to read. Interesting and relevant information for health care workers.
Details:
line 76 "Subjects lacking supplement data" any additional information here? Did these people refuse to answer questions related to supplement data? Do we anything about the reasons for missing data.
Figure 1 change "missed" to "missing"
line 120 - please provide SAS procedures used. It is my understanding that no statistical testing was done on changes over time. Please clarify in text. If time trends were assessed, do provide SAS code as an appendix.
Figures were all interesting but my retinas were tested as the font size is so small.
Line 214. This study was limited to participants with diabetes. No statements can be made inferring that people with diabetes are growing more on reliance on supplements than participants without diabetes.,
Line 293 personal bias of mine ... change "limits" to "prevents"
Author Response
Response to Reviewers 1
Comment 1: Overall, a pleasure to read. Interesting and relevant information for health care workers.
Response: Thank you for your encouraging comments.
Comment 2: line 76 "Subjects lacking supplement data" any additional information here? Did these people refuse to answer questions related to supplement data? Do we anything about the reasons for missing data.
Response:
Thank you for highlighting the issue with the missing data. Upon a detailed examination, we identified a mislabeling error in our flow chart regarding the missing data for dietary supplements and non-positive weights. We determined that the incidence of missing data for dietary supplements is exceedingly low throughout the study population (n=6). Unfortunately, NHANES did not specify the reasons for these data omissions. We have corrected and updated the flow chart in our manuscript.
Comment 3: Figure 1 change "missed" to "missing".
Response: Done, thanks.
Comment 4: line 120 - please provide SAS procedures used. It is my understanding that no statistical testing was done on changes over time. Please clarify in text. If time trends were assessed, do provide SAS code as an appendix.
Response: Thank you for your valuable suggestions. We employed logistic regression to assess trends in dietary supplement usage over time, which revealed a significant time trend (P < 0.0001). The SAS program for the entire paper is detailed in the supplementary materials attached to this paper. The corresponding SAS code is as follows:
data nh0920.merge41;
set nh0920.merge41;
If sddsrvyr in (6,7) then time=1;
else if sddsrvyr in (8,9) then time=2;
else if sddsrvyr in (66) then time=3;
run;
proc surveylogistic data=nh0920.merge41;
class time/ref= FIRST param=REF;
model DSD010(event='1') =time;
weight WTMEC12YR;
strata sdmvstra;
cluster sdmvpsu;
run;
Comment 5: Figures were all interesting but my retinas were tested as the font size is so small.
Response: We apologize for the inconvenience. We will attach the images in PDF format to allow for clearer scaling, as layout constraints prevent us from increasing the font size too much.
Comment 6: Line 214. This study was limited to participants with diabetes. No statements can be made inferring that people with diabetes are growing more on reliance on supplements than participants without diabetes.
Response: Thank you for your suggestion. We have incorporated this point into the limitations section of our revised manuscript as follows:
Furthermore, this study only included participants diagnosed with diabetes, and thus it is not appropriate to infer that individuals with diabetes are increasingly relying on supplements compared to those without diabetes.
Comment 7: Line 293 personal bias of mine ... change "limits" to "prevents"
Response: Done, thanks.
Reviewer 2 Report
Comments and Suggestions for Authors
Dear Authors:
Regarding the manuscript with title “Trends and Motivations in Dietary Supplement Use Among People with Diabetes: A Population Based Analysis from NHANES 2009-2020”, I have three major concerns. Also, several minor comments were addressed for the improvement of the quality of manuscript.
Major concerns:
Comment 1: Introduction needs more information to better contextualize the subject.
Comment 2: The chapter of Discussion needs more robust information regarding the interpretation and explanation of the findings. I will give some examples of how authors can improve this chapter.
2.1. - Lines 233-239: Authors have to better discuss for each demographic characteristic the possible reasons for the presented results. Also it will be of crucial importance to present data regarding U.S. population.
2.2. - Lines 247-248: Authors have to expose the reasons why folic acid, lutein and magnesium have potential benefits for insulin resistance and glycemic control.
2.3 - Lines 251-270. Authors must add references to the presented text.
2.4. - Authors have 4 lines on Discussion (lines 266-270) regarding the reasons for supplementation among diabetic patients and present 3 different figures regarding it (Figure 2c, 4 and 5). On Discussion authors only refer to Figure 2c. First, I suggest to better discuss the results found regading it (comparison with other studies and thoughtful explanations for results found) Besides, I question if Figure 4 and 5 are really needed among this manuscript? If authors considered relevant, they must discuss the results found.
Comment 3: Authors have to better align the purposes of the study with the Conclusions presented (both on Abstract ando n Conclusions chapter). Some importante topics regarding what is presented on manuscript: 1) prevalence of people with diabetes taking supplement; 2) differences on demographic and health-related characteristics between users vs. non users; 3) most common supplements used by diabetic patients; 4) efficacy evaluation by the comparison of the effect of different supplements on fasting glucose and HbA1c; 5) differences on demographic and health-related characteristics between Doctor-advised vs. Self-directed Supplement Use ;6) reasons for dietary supplement use.
Minor Comments:
Comment 1: Authors have to better expose the objectives of this manuscript, alligned with the presented data.
Comment 2: Line 57: “assess the motivations and reasons for supplement use”. Why authors refer to “motivations and reasons” and not only to “reasons”?
Comment 3: Line 66: I suggest authors to change “a nationwide, cross-sectional survey conducted over multiple years.” by “a nationwide, cross-sectional survey conducted over multiple years in United States.”
Comment 4: On Figure 1, authors have to add a legend regarding the meaning of the abbreviatures “FPG; “PG” and “HbA1c”.
Comment 5: On Figure 1, authors must have the exclusion criteria and not the inclusion criteria, since the number of the study participants decrease according to that.
Comment 6: On Figure 1, authors must change “PG≥200 11.1 mmol/L” by “PG ≥200 mg/dL (11.1 mmol/L)”
Comment 7: Line 80: I suggest authors to change “2.2. Definitions of diabetes” by “2.2. Definitions of diabetes and its complications”
Comment 8: Lines 138-140: “This trend was uniform across various baseline characteristics such as age, gender, and education level, indicating a broadening acceptance and utilization of supplements in this population”. I suggest authors to withdrawn the previous sentence as this information has already been mentioned on lines 141-143.
Comment 9: On Results, authors have to add a subchapter before presenting Figure 2 with reference to the results presented on Figure 2b and 2c.
Comment 10: On Figure 3, authors must refer the meaning of *; ** and *** on the legend. Also in the text refering to Figure 3 it will be importante to state in which variables it were noticed significant differences between groups.
Comment 11: Figure 2a and 2b must be presented after the text exposed on lines 184-190.
Comment 12: Authors have to correct the title of Table 3, as it does not represent the Table content.
Comment 13: Lines 266-268: “Finally, the reasons for dietary supplement use among diabetic patients, as shown in our study, are primarily centered around bone health, heart health, and immune enhancement.”. Given what is presented on Figure 2c, authors have also to refer to the reasons “Get more energy” and “supplement my diet.”
Author Response
Response to Reviewers 2
Comment 1: Introduction needs more information to better contextualize the subject.
Response: Thank you for your valuable feedback. We acknowledge the need for a more comprehensive introduction to better contextualize the subject of our study. We will expand this section to provide a clearer background and rationale for our research.
Comment 2: The chapter of Discussion needs more robust information regarding the interpretation and explanation of the findings. I will give some examples of how authors can improve this chapter.
Response: Thank you for your feedback regarding the Discussion chapter of our manuscript. We appreciate your specific suggestions on how to enhance the interpretation and explanation of our findings.
Comment 2.1: Lines 233-239: Authors have to better discuss for each demographic characteristic the possible reasons for the presented results. Also it will be of crucial importance to present data regarding U.S. population.
Response: Thank you for your valuable comments.
Here is the revised content of the paper:
Our study also highlights important demographic distinctions between supple-ment users and non-users. People with diabetes who use dietary supplements tend to be older, more educated, and are more likely to be female and white. According to pre-vious literature, it is estimated that 57.6% of Americans consume dietary supplements, with higher proportions observed among older adults and women(12). These charac-teristics are similar to those of our study's population. People with diabetes who use dietary supplements also exhibit better blood glucose control and report better overall health, with a lower prevalence of smoking. This demographic pattern suggests that certain groups may be more inclined to adopt supplementary treatments, possibly due to greater health awareness or access to health information(21, 22).
Comment 2.2: Lines 247-248: Authors have to expose the reasons why folic acid, lutein and magnesium have potential benefits for insulin resistance and glycemic control.
Response:
Thank you for your valuable comments. We have made the necessary additions and elaborations in the text.
Here is the revised content of the paper:
However, HbA1c levels were lower in subjects who took folic acid, lutein, and magne-sium. Consistent with previous studies(23-25), supplementation with lutein, folic acid, and magnesium is thought to have potential benefits for insulin resistance and glyce-mic control. Studies indicate that folic acid can enhance endothelial function and re-duce inflammation levels, both of which are crucial for improving insulin signaling(26, 27). Folic acid indirectly increases insulin sensitivity by lowering levels of homocyste-ine, an amino acid associated with insulin resistance(28). Lutein, an antioxidant, has been shown to reduce oxidative stress and inflammation, key drivers of insulin re-sistance(29). Additionally, lutein can improve the function of adipocytes, thereby aid-ing in the enhancement of insulin sensitivity(30). Magnesium supplementation directly boosts the activity of insulin receptors, thus improving insulin sensitivity and glycemic control(31). However, due to the cross-sectional design of our study, it is difficult to establish a causal relationship. Future prospective studies are needed to further iden-tify and confirm supplements that may have hypoglycemic effects.
Comment 2.3: Lines 251-270. Authors must add references to the presented text.
Response: Done,thanks.
Comment 2.4: Authors have 4 lines on Discussion (lines 266-270) regarding the reasons for supplementation among diabetic patients and present 3 different figures regarding it (Figure 2c, 4 and 5). On Discussion authors only refer to Figure 2c. First, I suggest to better discuss the results found regading it (comparison with other studies and thoughtful explanations for results found) Besides, I question if Figure 4 and 5 are really needed among this manuscript? If authors considered relevant, they must discuss the results found.
Response:
Thank you for your suggestion; we will discuss this in conjunction with the existing literature.To avoid redundancy, we have included Figures 4 and 5 as supplementary material in the revised manuscript.
Here is the revised content of the paper:
Finally, our study identifies the primary reasons for dietary supplement use among diabetic patients as improving overall health, maintaining wellness, supporting bone health, augmenting their diet, boosting energy levels, enhancing heart health, and strengthening immune function. These motivations appear closely linked to the common complications of diabetes, such as cardiovascular disease, bone demineralization, and weakened immunity. For example, cardiovascular issues, a frequent complication of diabetes, may explain why many patients prioritize heart health when selecting supplements(37). Similarly, the focus on immune function could be attributed to the increased susceptibility to infections observed in diabetic populations(38). Moreover, the diverse range of reasons for supplementation may also reflect varying patient perceptions about disease management and health optimization. While some patients may focus on addressing immediate health challenges, such as fatigue or immune support, others may be motivated by preventative strategies, aiming to reduce the long-term impact of diabetes-related complications. These patterns underscore the importance of personalized patient education.
Comment 3: Authors have to better align the purposes of the study with the Conclusions presented (both on Abstract ando n Conclusions chapter). Some importante topics regarding what is presented on manuscript: 1) prevalence of people with diabetes taking supplement; 2) differences on demographic and health-related characteristics between users vs. non users; 3) most common supplements used by diabetic patients; 4) efficacy evaluation by the comparison of the effect of different supplements on fasting glucose and HbA1c; 5) differences on demographic and health-related characteristics between Doctor-advised vs. Self-directed Supplement Use ;6) reasons for dietary supplement use.
Response:
Thank you for your constructive comments. We acknowledge the need to more closely align the objectives of our study with the conclusions presented in both the abstract and the conclusions chapter.
Here is the revised content of the paper:
Our analysis of NHANES data from 2009-2020 indicates an increasing trend in dietary supplement use among individuals with diabetes. This rise reflects a proactive approach by patients to enhance health outcomes and manage their condition more effectively. However, it also prompts important questions about how these supplements are integrated into conventional treatment regimens. Dietary supplement users are generally older, better educated, and exhibit superior glycemic control, indicating targeted use among those with greater health literacy. The most commonly used supplements include multivitamins, multimineral, vitamin D, calcium, zinc, vitamin C and fish oil, which are chosen for their general and diabetes-specific health benefits. Supplements such as folic acid and magnesium are beneficial for glycemic control, although further research is required to confirm these effects. Significantly, only 44.58% of supplement use is doctor-advised, with the majority being self-directed. This underlines the importance of healthcare providers playing an active role in guiding supplement use to ensure it is safe and effective. Healthcare providers should play a crucial role in advising patients on the appropriate use of dietary supplements, ensuring their use is grounded in sound scientific evidence and tailored to the individual’s overall health status and treatment goals. Further research is necessary to provide clearer guidance on the efficacy and safety of dietary supplements in the context of diabetes management.
Minor Comments:
Comment 1: Authors have to better expose the objectives of this manuscript, alligned with the presented data.
Response:Thank you for your valuable feedback. We will clarify and better articulate the objectives of our manuscript to ensure they are aligned with the presented data.
Here is the revised content of the paper:
To address this gap, we used latest data from the National Health and Nutrition Examination Survey (NHANES) to (1) describe the overall use of dietary supplements among people with diabetes, (2) compare the characteristics of supplement users and non-users, (3) identify the most commonly used supplements by those managing diabetes, (4) evaluate the efficacy of these supplements by comparing their effects on fasting glucose and HbA1c levels; (5) differences on demographic and health-related characteristics between Doc-tor-advised vs. Self-directed Supplement Use, and(6) assess the motivations and reasons for supplement use.
Comment 2: Line 57: “assess the motivations and reasons for supplement use”. Why authors refer to “motivations and reasons” and not only to “reasons”?
Response: We use both 'motivations' and 'reasons' to highlight the distinct yet complementary aspects influencing supplement use. 'Motivations' refers to the underlying psychological or emotional drives that prompt individuals to start using supplements, such as health promotion or disease prevention. 'Reasons', on the other hand, pertain to more immediate, rational explanations for supplement use, such as advice from a healthcare provider or specific health conditions. If you feel that 'reasons' is more appropriate, we will make adjustments in subsequent versions.
|
Motivations(To Use) and Reasons(For Use) for Using Dietary Supplements(Multiple choice) |
|
To: Build muscle Gain weight Get more energy Improve digestion Improve my overall health Maintain health (to stay healthy) Maintain healthy blood sugar level, diabetes Prevent colds, boost immune system Prevent health problems Supplement my diet (because I don’t get enough from food) For: Anemia, such as low iron Bone health, build strong bones, osteoporosis Eye health Good bowel/colon health Healthy Joints, arthritis Healthy skin, hair, and nails Heart health, cholesterol Kidney and bladder health, urinary tract health Liver health, detoxification, cleanse system Menopause, hot flashes Mental health Muscle related issues, muscle cramps Pregnancy/breastfeeding Prostate health Relaxation, decrease stress, improve sleep Teeth, prevent cavities Weight loss |
Comment 3: Line 66: I suggest authors to change “a nationwide, cross-sectional survey conducted over multiple years.” by “a nationwide, cross-sectional survey conducted over multiple years in United States.”
Response:Done, thanks.
Comment 4: On Figure 1, authors have to add a legend regarding the meaning of the abbreviatures “FPG; “PG” and “HbA1c”.
Response:Done, thanks.
Comment 5: On Figure 1, authors must have the exclusion criteria and not the inclusion criteria, since the number of the study participants decrease according to that.
Response:Done, thanks.
Comment 6: On Figure 1, authors must change “PG≥200 11.1 mmol/L” by “PG ≥200 mg/dL (11.1 mmol/L)”
Response:Done, thanks.
Comment 7: Line 80: I suggest authors to change “2.2. Definitions of diabetes” by “2.2. Definitions of diabetes and its complications”
Response:Done, thanks.
Comment 8: Lines 138-140: “This trend was uniform across various baseline characteristics such as age, gender, and education level, indicating a broadening acceptance and utilization of supplements in this population”. I suggest authors to withdrawn the previous sentence as this information has already been mentioned on lines 141-143.
Response:Done, thanks.
Comment 9: On Results, authors have to add a subchapter before presenting Figure 2 with reference to the results presented on Figure 2b and 2c.
Response:Thank you for your suggestion. We have split Figure 2 to ensure that the order of results presentation is consistent with the textual description.
Comment 10: On Figure 3, authors must refer the meaning of *; ** and *** on the legend. Also in the text refering to Figure 3 it will be importante to state in which variables it were noticed significant differences between groups.
Response:Done, thanks.
Comment 11: Figure 2a and 2b must be presented after the text exposed on lines 184-190.
Response:Thank you for your suggestion. We have split Figure 2 to ensure that the order of results presentation is consistent with the textual description.
Comment 12: Authors have to correct the title of Table 3, as it does not represent the Table content.
Response:Done, thanks.
Comment 13: Lines 266-268: “Finally, the reasons for dietary supplement use among diabetic patients, as shown in our study, are primarily centered around bone health, heart health, and immune enhancement.”. Given what is presented on Figure 2c, authors have also to refer to the reasons “Get more energy” and “supplement my diet.”
Response:Done, thanks.
Reviewer 3 Report
Comments and Suggestions for Authors
The study examined dietary supplement use among people with diabetes. Results indicated that 61.7% of individuals with diabetes use dietary supplements, with an increasing trend over time. These findings suggest a growing interest in alternative therapies among those with diabetes. The plagiarism index is low, and the results and statistics are presented thoroughly and well-organized in tables. The English text is well-written and clear, although there are some minor editorial issues where sentence structure could be improved for smoother readability. For example:
Abbreviations only need to be spelled out once, such as National Health and Nutrition Examination Survey (NHANES); afterward, NHANES alone is sufficient.
There is a double parenthesis here: (Weight (kg) / Height (m)²).
In tables, it’s clearer and more readable if there is a space between results.
An abbreviation list under the table (e.g., SE, NHANES, DM, CI, HDL, HBP, eGFR) would help ensure the table is understandable on its own.
The term people with diabetes is repeated several times. In some places, this could be replaced with simpler terms like patients or individuals to avoid redundancy.
In the phrase reflecting a growing trend, it would be more precise to use which reflects a growing trend, keeping in line with the previous sentence's timeframe.
Overall, the text is understandable and scientifically accurate.
Author Response
Response to Reviewers 3
Comment 1: The study examined dietary supplement use among people with diabetes. Results indicated that 61.7% of individuals with diabetes use dietary supplements, with an increasing trend over time. These findings suggest a growing interest in alternative therapies among those with diabetes. The plagiarism index is low, and the results and statistics are presented thoroughly and well-organized in tables.
Response: Thank you for your constructive feedback on our manuscript. We appreciate your recognition of our manuscript.
Comment 2: The English text is well-written and clear, although there are some minor editorial issues where sentence structure could be improved for smoother readability.
Response: Thank you for your positive comments on the clarity of our text and for highlighting areas where the sentence structure could be enhanced. We appreciate your attention to detail and will review the manuscript carefully to improve readability, addressing the minor editorial issues you have pointed out.
Comment 3: Abbreviations only need to be spelled out once, such as National Health and Nutrition Examination Survey (NHANES); afterward, NHANES alone is sufficient.
Response: Done,thanks.
Comment 4: There is a double parenthesis here: (Weight (kg) / Height (m)²).
Response: Done,thanks.
Comment 5: In tables, it’s clearer and more readable if there is a space between results.
Response: Thank you for your suggestion. We have added borders to the tables to enhance readability. If you still find that additional spacing is necessary, we will make further adjustments.
Comment 6: An abbreviation list under the table (e.g., SE, NHANES, DM, CI, HDL, HBP, eGFR) would help ensure the table is understandable on its own.
Response:Done,thanks.
Comment 7: The term people with diabetes is repeated several times. In some places, this could be replaced with simpler terms like patients or individuals to avoid redundancy.
Response:Done,thanks.
Comment 8:In the phrase reflecting a growing trend, it would be more precise to use which reflects a growing trend, keeping in line with the previous sentence's timeframe. Overall, the text is understandable and scientifically accurate.
Response:Done,thanks.
Round 2
Reviewer 2 Report
Comments and Suggestions for Authors
Authors give convincing responses to all my comments.